

# Continuous in-situ monitoring of nitrate concentration in soils – a key for groundwater protection from nitrate pollution

Elad Yeshno[1], Shlomi Arnon[2], Ofer Dahan[1]

[1]Department of Hydrology & Microbiology, Zuckerberg Institute for Water Research, Blaustein Institutes for
Desert Research, Ben Gurion University of the Negev, Israel
[2]Electrical and Computer Engineering Department, Ben-Gurion University of the Negev, Israel

*Correspondence to*: Elad Yeshno (Eladyes@post.bgu.ac.il)

**Abstract.** Lack of real-time information on nutrient availability in cultivated soils inherently leads to excess application of fertilizers in agriculture. As a result, nitrate, which is a soluble, stable and mobile component of fertilizers, leaches below the root zone through the unsaturated zone and eventually pollutes the groundwater and other related water resources. Rising nitrate concentration in aquifers is recognized as a worldwide environmental problem that contributes to water scarcity. Accordingly, developing technologies for continuous in-situ measurement of nitrate concentration in the soils are essential for optimizing fertilizer application and preventing water resource pollution by nitrate. Here we present a conceptual approach for a monitoring system that enables in-situ and continuous measurement of nitrate concentration in soil. The monitoring system is based on absorbance spectroscopy techniques for direct determination of nitrate concentration in soil porewater without pretreatment, such as filtration, dilution, or reagent supplementation. A new analytical procedure was developed to improve measurement accuracy while eliminating the typical measurement interference caused by soil dissolved organic carbon. The analytical procedure was tested at four field sites over 2 years and proved to be an effective tool for nitrate analysis in untreated soil. A soil nitrate-monitoring apparatus, combining specially designed optical flow cells with soil porewater-sampling units, enabled for the first time, real-time continuous measurement of nitrate concentration in the soil. The system provided outstanding and explicit data revealing the complexities of the temporal variations in soil nitrate concentrations in response to irrigation cycles and fertilizer-application pattern. Such real-time measurements of soil nitrate levels are crucial for optimizing fertilizer application to increase agricultural yield while reducing the potential threat of groundwater contamination by down-leaching of nitrate from the soil.



## 1. Introduction

Pollution of water resources by nitrate from agricultural sources is one of the main reasons for freshwater disqualification worldwide (Jin et al., 2012; Liu et al., 2005; Orban et al., 2010; Thorburn et al., 2003). The US Environmental Protection Agency (EPA) regards nitrate contamination in groundwater as an event requiring immediate action (EPA US and Office of Water, 1994). In addition, a Nitrates Directive has been established by the European Community to prevent water pollution by nitrate (European Community, 1991). In Israel, nitrate has contributed more than any other contaminant to water well shutdown events in the present century (Elhanany, 2009).

Water resource pollution by nitrate seems to be primarily caused by excessive application of agricultural fertilizers (Kourakos et al., 2012; Liao et al., 2012; Osenbruck et al., 2006). Nitrate concentration in soil porewater often changes rapidly, on a time scale of hours to days (Dahan et al., 2014). These rapid changes are dictated by irrigation/precipitation pattern, fertilization and cultivation methods, plant uptake, and natural soil biochemical processes (Oren et al., 2004; Thompson et al., 2007; Vázquez et al., 2006). Soil nitrate concentration is commonly estimated through measurement of soil porewater samples, which are obtained using a suction cup or soil sample extraction (Abdulkareem et al., 2015; Dahan et al., 2009; Evett and Parkin, 2005). The porewater sample or soil sample extract is then analyzed for nitrate by standard laboratory procedures, or with special kits for quick analysis in the field (Liebig et al., 1996). These measurement methods are not in line with the time scale of N-fertilizer mobilization, consumption and transformation dynamics in agricultural soils. Since there are as yet no "on-shelf" technical means for real-time continuous measurement of nutrient concentrations in the soil, farmers tend to apply an excess mass of N-fertilizer as common practice. The direct outcome is a continuous flux of nitrate from the root zone, through the unsaturated zone, to the groundwater (Burow et al., 2010; Fisher and Healy, 2008; Kurtzman et al., 2013; Oren et al., 2004; Scanlon et al., 2007). Although governments, environmentalists, and agricultural sectors are continuously seeking new ways to limit nitrate pollution, in the absence of monitoring tools capable of providing online, real-time, continuous measurements of nitrate in the soil, reducing groundwater nitrate pollution will remain out of reach.

Today, two main technologies are used for real-time analysis of nitrate in water samples: optical dip probes, based on ultraviolet (UV) absorbance spectroscopy, and ion-selective electrode (ISE) dip probes (De Marco et al., 2007). However, these methods are designed for water samples and not for direct application in soil. Nitrate analysis by UV absorbance spectroscopy is a common technique that has been implemented for several decades (Meyerstein and Treinin, 1961; Moorcroft, 2001), based on the principle that when electromagnetic energy, such as UV light, propagates through aqueous samples, a fraction of that energy can be transferred to some of the dissolved ions through the transition of electrons between different energy levels (West, 2014). The intensity of the energy absorbed by the ions is proportional to their concentration in the solution. UV absorbance spectroscopy has been found highly effective for measuring nitrate concentration directly from aqueous samples, as it does not require any addition of reagents, thus making it less time-consuming and more reliable than other spectral techniques (Ferree and Shannon, 2001). This method is considered more stable and robust than the ISE probe method because UV absorbance spectroscopy is not sensitive to changes in temperature, pH or salinity of the water solution (Edwards et al., 2001). Several methods for nitrate estimation in aqueous solution by absorbance spectroscopy techniques have been practiced for several decades. Tuli et al. (2009) demonstrated the ability to




measure nitrate at 235 nm. Moo et al. (2016) showed nitrate measurements at 302 nm, and (Michael et al., 2017) measured nitrate concentration at 200 and 220 nm.

The simplicity and robustness of UV absorbance spectroscopy for measuring nitrate concentration in water samples make it potentially applicable for in-situ application in soil. Tuli et al. (2009) suggested an in-situ

method for monitoring nitrate in saturated media by measuring the nitrate concentration in a solution held inside a stainless-steel porous cup. In their proposed method, the porous cup is filled with deionized water and then lowered into a reservoir containing nitrate solution. An optical dip probe is then placed inside the porous cup to perform the spectral analyses. The suggested setup has shown great potential for in-situ monitoring of nitrate concentration. However, the time required for the solution inside the porous cup to reach equilibrium with the

surrounding solution (up to 60 h) negates the use of this apparatus for measuring nitrate concentration at high time resolution when placed in the soil. Moreover, the equilibrium times are expected to become significantly longer when the measurement is conducted in unsaturated soils (Riga and Charpentier, 1998).

Although UV absorbance spectroscopy for nitrate analysis is very common, it has some limitations when applied to natural water samples, which contain a variable concentration of dissolved organic carbon (DOC).

Shaw et al. (2014) studied the possible interference in UV absorbance spectroscopy for nitrate analyses by the different ions that are commonly found in water samples originated from natural sources. They showed that the main interference is caused by DOC, with the nitrate absorbance signal being completely quenched above 50 ppm DOC (Shaw et al., 2014). As a result, absorption-signal masking by DOC, which is commonly found in agricultural soils, can prevent the use of UV absorbance-based methods for nitrate evaluation in water samples

(Fig. 1).

The interference caused by DOC can often be reduced by applying the dual-wavelength correction scheme (Armstrong, 1963). In this method, nitrate concentration is estimated through the value of twice the absorbance intensity at 275 nm deducted from the absorbance value at 220 nm. However, this method can only be used when the absorbance intensity at 275 nm is lower than 5% of the absorbance measured at 220 nm.

Additional method that accounts for DOC interference is second-derivative spectroscopy (Causse et al., 2017; Crumpton et al., 1992; Ferree and Shannon, 2001; Simal et al., 1985). In this technique, nitrate is measured at the ~224 nm absorbance peak of the second-derivative spectrum. Ferree and Shannon. (2001) reported the ability to measure nitrate concentration in water samples from wetlands and treated wastewater which contained up to 77 ppm DOC. However, a primary condition of the analyses is that the samples be at a concentration lower than 44.3

ppm nitrate. Yet, since nitrate concentration in cultivated and fertilized soils may vary through a wide range of tens to thousands of parts per million, following fertilization cycles, a dilution of the samples would be necessary to measure nitrate by the second derivative spectroscopy technique, Thus, making this method less applicable for continuous in-situ measurement.

In this paper, we present a novel technique for measuring nitrate concentration in soil porewater based

on UV absorbance spectroscopy technique. The method is based on scanning the absorption spectrum and identifying an optimal wavelength for repetitive measurements of nitrate concentration in the soil porewater that overcomes the typical analytical interference by DOC. The analytical procedure is combined with a novel approach that enables continuous measurement of the UV absorption spectrum in an optical flow cell connected to a porous interface to enable continuous in-situ monitoring of nitrate concentration in the soil. We believe that



the proposed monitoring technology could open a new avenue for precision fertilization and optimization of crop production, while reducing the risks associated with nitrate pollution of groundwater.

## 2. Material and methods

In order to develop an analytical procedure capable to carry continuous measurement of nitrate concentration in the soil, porewater samples were collected from various typical cultivated sites and analyzed for
their chemical composition and spectral characteristics. The analytical spectral procedure developed on the basis of the spectral characteristics of the soil porewater was then tested in soil columns, which were equipped with a specially designed optical setup for continuous measurement of nitrate concentration in the soil.

### 2.1 Selected agricultural sites

Four typical agricultural fields were selected: (i) organic and (ii) conventional greenhouses for vegetable
crops, (iii) an open crop field with rotating seasonal crops, and (iv) a citrus orchard. All sites were located in the agricultural area of Israel's coastal plain. The porewater samples were collected by vadose zone-monitoring systems (VMSs) that have been operating at these sites continuously for more than 9 years. The VMS includes a porewater sampler that is permanently installed in the unsaturated zone under the cultivated fields. Accordingly, variations in the chemical characteristics of the soil porewater may be detected continuously at the same spot in
the subsurface over many years. A detailed description of the VMSs at each site can be found in Dahan et al. (2014) and Turkeltaub et al. (2014, 2015, 2016), and in section S1. Additional information on the research site locations, crop types, and irrigation and fertilization regimes can be found in section S2. The porewater-sampling ports at each site are distributed at various depths, ranging from 1 to 21 m (Table S3). In this study, soil water samples were collected in four sampling campaigns: (i) August 2015, (ii) September 2015, (iii) January 2017 and
(iv) February 2017.

### 2.2 Spectral and chemical characteristics of the soil porewater

Samples were analysed for nitrate concentration with a Dionex ICS 5000 ion chromatograph and the Analytik Jena TOC, DOC, TN, DN multi N/C 2100s TOC/TN analyzer for DOC and total nitrogen (TN) concentration. Spectral analyses of the samples were performed with a Thermo Scientific Evolution 201/220
Desktop laboratory spectrophotometer. Double-distilled water (DDW) was used as a reference/baseline for the analyses. The samples were held in a standard 5-mL quartz cuvette with an optical path of 10 mm and were scanned over a broad spectrum of 190–1000 nm. The analytical procedure for UV spectral analysis of nitrate concentration in porewater samples usually requires colloid filtration, dilution and sometimes spiking with the target constituent or supplementary reagents. However, since the purpose of this study was to develop an analytical
protocol that enables in-situ measurement of nitrate concentration through spectral analyses of the soil porewater, the samples were analyzed without any additional preparation (i.e., dilution or filtration).

To validate our suggested method's resistance to measurement drift, which may occur in response to changes in the solution chemical matrix, a second spectral analysis was performed. This analysis was carried out in a Spark 10M multimode microplate reader spectrophotometer at wavelengths of 200 to 1000 nm. The accuracy of the





suggested method was determined by the correlation strength, or coefficient of determination ($R^2$), between the absorbance intensity values and nitrate concentration values measured by the ion chromatograph. Absorbance was defined by the Lambert–Beer equation (Eq. (1)):

$$\text{Absorbance} = -\log_{10}\frac{I}{I_0} \tag{1}$$

where I is the light intensity after passing through the examined solution and $I_0$ is the light intensity after passing

through a reference sample (blank).

### 2.3 Optical flow cell

       In order to enable continuous in-situ measurement of nitrate concentration in the soil, a monitoring concept was developed in which the spectral absorption of the soil porewater is measured in an optical flow cell (Fig. 2) (a patent is pending on the methodology described in this manuscript). The optical setup consists of a UV

lamp and UV–VIS spectrometer, designed to measure transmission and absorbance between 190 and 850 nm. A special feature in SpectroWiz (StellarNet software) was used to prevent possible measurement drift. A StellarNet SL3 deuterium light source was used as continuous-wave UV light source. The spectrometer and UV lamp were connected to a flow cell using optical fibers and collimating lenses. The optical flow cell was connected at one end to a customized suction cup, which enables continuous sampling of the soil porewater under a low flow rate

(a few milliliters per hour). At the other end, the flow cell was connected to a sampling cell. Charging the sampling cell with low pressure draws a continuous flux of porewater from the soil through the optical flow cell to the sampling cell. The system is designed to function under a small dead volume (4–6 mL) by reducing the suction cup's inner volume and using small-diameter tubing (inner diameter 1.6 mm). Porewater solution that flows from the suction cup through the optical cell accumulates in the sampling cell, and is used later to determine nitrate

concentration by standard laboratory procedure.

### 2.4 Column experiment

       The monitoring system for continuous measurement of nitrate concentration in the soil was tested in two sets of column experiments. The first was conducted to test the ability of the optical setup to measure nitrate concentration in the soil under controlled conditions. In this experiment, 18 L of clean (low organic matter) quartz

sand was packed in a 50-cm long column. Two identical customized suction cups and one water-content sensor (TDT, Acclima) were placed at a depth of 22 cm in the soil column. One of the suction cups was connected to the flow cell, and the other directly to its sampling cell (Fig. 2). The column was irrigated daily with 1 L fresh tap water, where one of the irrigation cycles was enriched with 1000 ppm nitrate (as $KNO_3$). In this experiment, nitrate concentration of the soil porewater was measured continuously using absorption spectroscopy technique in the

optical flow cell and compared to the concentration in the porewater samples that were accumulated in the two sampling cells and in the column drainage. The second experiment was conducted using agricultural soils in three soil columns packed with fine sandy soil, dark clay soil and fine sand soil mixed with 10% commercial compost, respectively. The experiments were conducted in all three columns under similar irrigation, fertilization and monitoring setups (Table S4).




## 3. Results and discussion

### 3.1 UV absorption characteristics of agricultural soil porewater

The UV absorption spectra of soil porewater were measured in water samples obtained from four representative agricultural fields: organic and conventional greenhouses, a citrus orchard and an open crop field. The porewater samples were obtained from each site by the VMS from 6–8 sampling points distributed vertically and laterally under each site (S3). Note that the VMS sampling ports are permanently installed at the site and therefore enable repeat sampling from the exact locations for many years, while the agricultural activity on land surface remains undisturbed (Turkeltaub et al., 2015, 2016).

All water samples were first analyzed for nitrate and DOC concentration by standard laboratory protocols. The porewater samples were then examined for absorption at a few specific wavelengths that have been previously suggested for direct nitrate measurement in untreated soil water: (i) 302 nm (Moo et al., 2016), (ii) 235 nm (Shaw et al., 2014; Tuli et al., 2009)), and (iii) where the absorbance used for calibration equals the absorbance at 220 nm after subtraction of twice the absorbance at 275 nm (hereafter 220/275 nm) (Armstrong, 1963). An additional measurement at 220 nm, as suggested by (Michael et al., 2017), was also carried out, but there was no significant difference in absorption characteristics compared to the 220/275 nm method. Therefore, the data from this test are not presented. Note, that since the porewater samples were obtained from the soil through a porous interface (air entry pressure 1–1.5 bar), no additional filtration or other treatment was performed. Moreover, since the overall goal of this study was to develop a method for direct measurement of nitrate concentration in the soil, the absorption spectra were measured in untreated water, as obtained directly from the soil.

Nitrate concentration plotted against absorbance intensity at the selected wavelengths for all sites showed inconsistencies between the two (Fig. 3). At 302 nm (Fig. 3a), a reasonable correlation between absorbance intensity and nitrate concentration was obtained for the open crop field ($R^2 = 0.99$) and conventional greenhouse ($R^2 = 0.95$), whereas poor correlations were obtained for the other two fields: organic greenhouse ($R^2 = 0.39$) and citrus orchard ($R^2 = 0.49$). Partial improvement was achieved at 235 nm (Fig. 3b), with $R^2$ values of 0.97, 0.91 and 0.98 for the organic greenhouse, open field crop and conventional greenhouse, respectively. However, a poor correlation was obtained for water samples from the orchard ($R^2 = 0.71$). Moreover, a close inspection of the absorbance intensity of water samples from the open crop field showed a strong shift in absorbance intensity at nitrate concentrations exceeding 1000 ppm. This phenomenon was observed in repeat analyses of additional water samples (Fig. S5). With the 220/275 nm method (Fig. 3c), poor correlations between absorbance intensity and nitrate concentration were observed at most sites ($R^2 = 0.39$, 0.09, 0.75 for organic greenhouse, open field crop and conventional greenhouse, respectively); however, for the orchard site, the correlation was improved compared to the other methods, reaching $R^2 = 0.9$. Note that one of the porewater samples from the organic greenhouse (from 13.3 m with 171.36 ppm nitrate) did not meet the requirements of the 220/275 nm absorbance ratio and is therefore not included in Fig. 3c. None of the methods based on specified wavelengths seemed robust enough for direct analysis of untreated soil water obtained from various fields with different soils.

Several reasons could account for the observed mismatch between absorbance intensity and nitrate concentration at the various sites. At short wavelengths, such as 220 nm, the absorbance intensity is typically very high (Fig. 1) and therefore, the measurement is very sensitive to low nitrate concentrations. At high nitrate concentrations, however, absorption saturation occurs, and the absorbance intensity is no longer indicative of


increased concentrations. Accordingly, in agricultural soils, where nitrate concentration may vary from tens to thousands of parts per million, as demonstrated in the water samples obtained from sites used for this research, the shorter wavelengths are less applicable for direct analysis (i.e., the samples need to be diluted). This explains the low correlation found for 220/275 nm and the low sensitivity to high concentration at 235 nm. The 300 nm region is typically characterized by low absorption rates for nitrate (Fig. 1), thereby reducing the potential for

signal saturation. As such, it is more ideal for measuring nitrate at high concentrations. Our measurements at 302 nm were insensitive to the low nitrate concentrations (49.7–75.4 ppm) at the orchard site. Furthermore, significant mismatch was observed for the organic greenhouse, even though the nitrate concentration at this site was relatively high, ranging from 171 to 520 ppm (Fig. 3a). This mismatch was expressed as increasing absorbance intensity values, regardless of the nitrate concentration. The main reason for the increased absorption could be attributed to

signal masking as a result of the presence of DOC, which is commonly found in agricultural soil porewater (Jones and Willett, 2006; Kalbitz et al., 2000). Nevertheless, a closer look at the absorption pattern showed that different soils may have proper calibration curves for nitrate concentration at different wavelengths, allowing for the possibility of adopting a unique wavelength for each site.

### 3.2. DOC and nitrate concentrations impact the UV absorption spectra

The soil porewater samples obtained in this study contained DOC concentrations ranging from 0 to 14 ppm. The nitrate absorption spectra of the porewater samples were therefore susceptible to signal masking as a result of the strong absorption of UV light by DOC molecules. However, a deeper analysis of the absorption spectra revealed a complex nature of the relationship between DOC concentration and absorbance intensity in the UV range. This complex pattern was best seen in the spectral characteristics of porewater samples obtained from

the organic greenhouse, where composted dairy and poultry manure is used as the main fertilizer, enriching the soil with excessive concentrations of organic matter. The absorption spectrum of porewater samples obtained from various depths under the organic greenhouse showed the highest absorbance intensity for samples from cells located at 1.3 m (Fig. 4a), despite having the lowest nitrate concentration in the sample batch (Fig. 4b). Although the high absorbance values might be attributed to the presence of DOC, these water samples did not have the

highest DOC concentration. On the other hand, the water sample from 13.3 m, which did have the highest DOC concentration of the concurrent batch (Fig. 4b), showed the lowest absorbance intensity value (Fig. 4a). This peculiar behavior was found consistently in subsequent sampling campaigns (Fig. S6). Thus, it could be concluded that DOC absorption characteristics are not impacted solely by the overall DOC concentration but also influenced by the specific characteristics of the various organic compounds composing the overall DOC. Accordingly,

different soils at different sites could potentially be characterized by different organic compounds in their specific DOC "soup", which could therefore have its own typical absorption spectrum.

### 3.3. Nitrate vs. DOC UV absorption spectrum

The attempts to measure nitrate concentration at a specific wavelength (302, 235 and 220/275nm) showed inconsistencies between the absorption characteristics and nitrate concentration, attributed to absorption saturation

and the presence of DOC. However, DOC concentration was not always correlated with absorbance intensity. As a result, a new approach was adopted to better assess the effect of nitrate and DOC concentrations on the



absorption spectra. In this approach, the coefficient of determination ($R^2$) between a set of nitrate/DOC concentration vectors and their corresponding absorbance intensity vectors was calculated for the entire spectrum (Fig. 5, Table 1 and Fig. S7).

The coefficient of determination ($R^2$) vs. wavelength, for both nitrate and DOC concentrations, are shown in Fig. 5a for the open crop field and Fig. 5b for the citrus orchard samples. The $R^2$ values for nitrate in the crop field shows an increase at 225 nm, reaching a plateau ($R^2 > 0.99$) between 235 and 250 nm. They then decreased to a minimum value of 0.57 at 264 nm, and rose again to a second, high-value plateau (>0.9) between 290 and 320 nm. However, the $R^2$ pattern for the DOC concentrations in the crop field differed from that for nitrate. In

some sections (220–235 and 225–360 nm), the trends were positively correlated, whereas in others (250–325 nm) they were either negatively correlated or not correlated (Fig. 5a). Unlike the case of the open crop field, where two distinct high $R^2$ value plateaus were visible, analysis of the citrus orchard $R^2$ values showed only a narrow area with high $R^2$ values between the wavelengths of 220 and 230 nm. Here, the high $R^2$ values (>0.8) were only reached at 220–235 nm, whereas for the rest of the spectrum, the correlation was very poor (<0.4) (Fig. 5b). On

the other hand, at this site, $R^2$ values for the DOC remained very low (<0.3) over the entire spectrum. A similar $R^2$ vs. wavelength analysis was carried out for the other fields and the trend in $R^2$ for each field seemed to show unique behavior (S7).

        The wavelength regions with high $R^2$ values showed a higher correlation between the targeted chemical concentration (nitrate or DOC) and absorbance values. Thus, absorbance values in those areas had greater potential

for measuring the targeted constituent's concentration. For example, in the open field crop, the areas of the two distinct high $R^2$ plateaus (Fig. 5a) hold high potential for measuring nitrate concentrations in soil porewater collected from that field. Between the two sections of high correlation to nitrate concentration, at around 267 nm, absorbance intensity values were correlated with DOC concentration, meaning that this area of the spectrum is expected to have a high DOC masking effect. These characteristics were unique to the open crop field. In the

citrus orchard (Fig. 5b), for example, the data series associated with nitrate concentration presents high potential for estimating nitrate concentration at wavelengths between 220 and 230 nm. Moreover, the low $R^2$ values for the DOC curve suggest that the DOC chemical composition in the citrus orchard porewater samples does not have much effect on the UV absorbance absorption spectrum over a greater section of the spectrum.

        Although DOC concentration in porewater at the different sites in this study was rather similar (Table

S8), the DOC impact on the absorption spectrum was very different at each site. It was assumed that these variations are due to the composition of the various organic molecules making up the DOC in the different fields. DOC is a general term folding thousands of different organic molecules within it. Accordingly, the specific chemical composition of the DOC may be affected by various factors, such as differences in soil type, crop type, differences in the applied fertilizers and local climate (Kalbitz et al., 2000). For example, regardless of the

proximity between DOC concentration values in the citrus orchard and the open crop field, the presence of DOC did not cause similar interference in the spectral analyses of the porewater at those sites. In fact, in the crop field, where DOC concentrations were slightly lower than those in the citrus orchard, the presence of DOC had a much higher impact on the absorption spectra of the porewater samples taken from the crop field compared to samples taken from the citrus orchard. Nevertheless, every field site can be characterized by wavelength regions that have

greater potential for measuring nitrate concentration, and those that might be more susceptible to interference by DOC or other constituents in the solution (Fig. 5). This phenomenon opens the way to a new concept, whereby a




wavelength can be determined that is uniquely suited to measuring nitrate in each field while avoiding possible interference related to other natural water constituents, such as DOC.

### 3.4. Determination of optimal wavelength for site-specific calibration

The observed variations in the coefficients of determination for nitrate and DOC concentrations at different wavelengths (Fig. 5) led to the adoption of an innovative strategy for analyzing nitrate concentration by absorbance spectroscopy. The new analytical procedure was designed to overcome the measurement inconsistencies associated with estimations of nitrate concentration using absorbance spectroscopy methods with a fixed wavelength (Fig. 3).

A two-step procedure was used to determine the optimal wavelength for nitrate concentration measurements in soil porewater samples at specific sites. The first step consisted of creating a set of candidate wavelengths that show high potential for measuring nitrate concentration. This was achieved by plotting the $R^2$ values of absorbance intensities of known nitrate concentrations vs. wavelength (Fig. 5). The candidate wavelengths were then screened to satisfy two requirements:

(i) $R^2$ test: an initial screening of the wavelength range was performed by setting a threshold value that is within 98% of the maximum $R^2$ value in the tested batch (Fig. 6). Wavelengths showing $R^2$ values below that threshold were rejected, while the wavelengths displaying $R^2$ values above the threshold were used to form a set of candidate wavelengths for a site-specific calibration equation. In this example, $R^2_{max} = 0.9953$, so the $R^2$ threshold value was set to $R^2_{98\%} = 0.9753$.

(ii) Variance ($\sigma^2$): calibration curves created for various wavelengths, for example 238 nm and 300 nm for the open crop field, showed high $R^2$ values of 0.9792 and 0.9869, respectively. Either wavelength could be used to set up a suitable calibration curve. However, the calibration curve related to 300 nm had a much steeper slope, indicating lower variance ($\sigma^2$) compared to the calibration curve related to 238 nm (Fig. 7). The slope of the calibration curve, which reflects $\sigma^2$, has a high impact on the sensitivity of the analyses to measurement errors.

Accordingly, with a sharp slope calibration curve (low $\sigma^2$), as in the case of 300 nm for the crop field, a slight variation in absorbance intensity will result in greater errors in the estimated nitrate concentration values. Hence, the strength of the calibration curve cannot be estimated solely by the coefficient of determination ($R^2$). Accordingly, the second parameter, variance ($\sigma^2$), which is derived from the measured absorbance intensity values, was used to quantify the sensitivity of a calibration curve to measurement errors.

The site-specific optimal wavelength was determined by combining the $R^2$ and $\sigma^2$ values for each wavelength; the square root of the sum of the two criteria's values Eq. (2) was calculated for those wavelengths that have $R^2$ values above the set threshold. Figure 6 shows that at a wavelength of 238 nm, a peak point on the curve emerges, indicating that it is the most suitable wavelength for spectral analysis of nitrate concentration for this particular site (open crop field).

$Combined\ criteria = \sqrt{R^2 + \sigma^2}$                                       (2)

        Application of this procedure to determine the optimal wavelengths for all fields used in this study enabled establishing a specific calibration curve for each site. Plotting the nitrate concentration as obtained by ion chromatograph against absorbance intensity at multiple wavelengths (organic greenhouse at 231 nm and $R^2 = 0.99$, open crop field at 238 nm and $R^2 = 0.99$, conventional greenhouse at 234 nm and $R^2 = 0.99$, and citrus





orchard at 223 nm and $R^2 = 0.98$) showed very high correlations. In this case, each of the fields was successfully assigned to an individual calibration curve, generated by the most suitable wavelength for that specific site. Figure 6 shows information for the open crop field station; further information for the two-step procedure's application to the other field stations is presented in section S9. Note that the poorly correlated data in Fig. 3 and the highly correlated data in Fig. 8 were produced from same absorption spectra of the same water samples. The only difference is that the data in Fig. 3 were created by application of fixed wavelengths of known methods, whereas the highly correlated data in Fig. 8 were created on the basis of an analytical procedure that searches for a site-specific optimal wavelength.

### 3.5. Stability and consistency of the specific calibration curves

As already noted, the main goal of this study was to develop a robust analytical procedure and technical means for continuous in-situ measurement of nitrate concentrations in shallow soil and the deep unsaturated zone, to optimize fertilizer application and prevent groundwater pollution. Accordingly, the robustness of the method is primarily dependent on the temporal stability of the site-specific calibration equations. There are two main reasons for calibration drift: (i) drift in the optical apparatus due to light source degradation or intensity fluctuations and (ii) changes in the porewater solution matrix chemical composition, which might lead to absorbance-signal masking or other interference patterns in the spectral analyses.

To validate the temporal stability of the site-specific calibration equation obtained by the spectral analytical procedure, additional porewater samples were collected and analyzed at different times (September 2015, January and February 2017). All water samples from all sampling campaigns were tested with the calibration curves and optimal wavelengths set for the samples from August 2015. A comparison of the nitrate concentrations predicted on the basis of the site-specific calibration curves from August 2015 to the observed nitrate concentrations showed a good correlation with general $R^2 > 0.9$ (Fig. 9). Accordingly, it is suggested that the initial calibration equation which was determined by the spectral analytical procedure 2 years earlier (2015) was still valid for nitrate concentration estimations, regardless of the changes in agricultural activity between growing seasons. It may therefore be concluded that establishment of a site-specific calibration curve that is based on the adoption of a site-specific wavelength can be used for long-duration monitoring of nitrate in soil porewater, as long as stability of the UV light source is maintained.

### 3.6. Real-time monitoring of nitrate concentration in the soil

### 3.6.1. Column experiments

Application of a site-specific calibration curve based on a determination of optimal wavelength enabled measuring nitrate concentration in soil porewater from agricultural fields. However, measuring nitrate concentration in porewater in real time requires applying absorbance spectroscopy techniques directly to the soil porewater. The monitoring concept proposed here includes continuous measurement of the UV spectrum absorption in an optical flow cell. In this setup, a continuous flow of soil porewater is obtained at a very low rate (a few milliliters per hour) from the soil through the optical flow cell (Fig. 2). This monitoring concept, which is referred to here as an optical nitrate sensor, was tested in two types of column experiments: (i) controlled column



experiment - conducted in clean sandy soil with very low organic matter content, designed to test the performance of the monitoring setup without measurement interference caused by DOC, and (ii) agricultural soils - three column experiments were conducted in three agricultural soils-sandy soil, dark clay soil, and sandy soil-mixed with commercial compost. This experiment was designed to test the optical setup's ability to measure nitrate in natural cultivated soils containing natural DOC. All of the column experiments were conducted through daily irrigation cycles, where one of the irrigation cycles was replaced with enriched nitrate solution (1000 ppm).

### 3.6.2. Nitrate breakthrough curve during the controlled column experiment

Nitrate breakthrough in the soil column was established by continuous measurement of nitrate concentration, as obtained from the UV absorption spectrum in the optical flow cell, and by daily measurement of nitrate concentration (by a laboratory method) in water samples obtained from two suction lysimeters and from the column drainage (Fig. 10). Daily sampling of the suction lysimeters and drainage exhibited the expected breakthrough curve, with the drainage showing delayed breakthrough and a lower maximum concentration compared to the two lysimeters, which were practically identical. Ultimately, the continuous measurement of nitrate concentration in the soil provided outstanding explicit data on the complexity of its temporal variation in the soil. In general, the nitrate breakthrough curve generated by the optical nitrate sensor was fairly consistent, showing similar concentration and variation trends. Moreover, the data obtained by the optical nitrate sensor revealed the real complexities of the changes in nitrate concentration with respect to the dynamics of water percolation in response to the irrigation events. The breakthrough curve obtained by the optical nitrate sensor exhibited a higher maximum concentration than those obtained by the lysimeters. This, however, might be attributed to the obvious fact that the samples being collected by the lysimeter represent daily averaged values of a cumulative sample, while the optical nitrate sensor provides continuous online measurements of the soil porewater. Sampling the soil solution as a cumulative sample, as with the suction lysimeters, will miss the temporal fluctuations in soil nitrate concentration. A closer look at the breakthrough curve structure for the high-time-resolution measurement of nitrate concentration in the soil porewater reveals rapid changes in nitrate concentration following irrigation and soil-wetting cycles (Fig. 10). The relationship between the irrigation events and the rapid changes in nitrate concentration is directly attributed to mechanisms controlling water flow and solute transport within the porous domain. Obviously, this phenomenon is of great importance and relevance to the soil and hydrological sciences, as regards solute and contaminant transport. However, further analysis of this phenomenon was beyond the scope of the presented study.

### 3.6.3. Real-time measurement of nitrate concentration in agricultural soil

Following the controlled column experiment, which proved the ability to carry out continuous spectral absorption measurements in soil porewater, and following the analytical procedure that enabled developing a site-



specific calibration curve, a column experiment was performed with agricultural soils. These experiments were conducted under conditions similar to those of the controlled experiment, where irrigation was applied on a daily base with one of the cycles being replaced with a nitrate-enriched solution (1000 ppm). The breakthrough curves of nitrate obtained by the optical nitrate sensor were then compared with those from water samples obtained by suction lysimeters (Fig. 11). The breakthrough curves obtained from the column experiments in all soils were based on the spectral analytical procedure for determining optimal wavelengths for measuring nitrate concentration. Accordingly, the optimal wavelengths were set to 231.82 nm for the dark clay soil, 230.66 nm for the sandy soil and 223.86 nm for the sandy soil mixed with compost.

Outstanding similarity was found between the optical sensor-calculated data and the nitrate concentrations from the laboratory analysis. Accordingly, the correlation coefficients for the regression of the physically vs. optically obtained data showed high values: $R^2_{controlled\ column} = 0.91$, $R^2_{sandy\ soil} = 0.94$, $R^2_{sandy\ soil\ +\ compost} = 0.87$ and $R^2_{clay\ soil} = 0.92$. Moreover, the automatically obtained, high-resolution real-time measurements provided the first observation of rapid changes in nitrate concentration correlated to the irrigation patterns. Such observations could not have been made in the agricultural environment, where soil solution sampling can be practically performed only at much longer time intervals, or even under the exclusive conditions available for a controlled scientific experiment, where only daily sampling of the suction lysimeter is possible.

### 4. Conclusion

The lack of online in-situ instrumentation for monitoring nutrient availability in the soil often results in excess application of nitrogen-based fertilizers. Consequent nitrate leaching from the root zone to the deep unsaturated zone can result in severe groundwater pollution. Our newly developed optical sensor enables, for the first time, continuous in-situ measurement of nitrate concentrations in the soil. The new monitoring concept was based on the application of UV absorption techniques to porewater obtained continuously from the soil. To avoid spectral interference by DOC, an analytical procedure that scans the entire UV spectrum was used to determine a site-specific optimal wavelength and calibration equation for nitrate concentration measurements. Applying the analytical procedure to the soil porewater from the different agricultural sites revealed that each site can be characterized by a single optimal wavelength that enables repetitive nitrate measurements. The spectral analysis procedure was then combined with an optical flow cell to form an optical soil nitrate sensor (patent pending). The sensor was tested in a series of column experiments showing outstanding ability to measure nitrate concentration accurately at high time resolution in all tested soils. This work provides a scientific basis for the development of a nitrate-monitoring system that that would be capable of providing high-resolution in-situ nitrate concentration measurements in soils, while minimizing possible interference from the presence of DOC. We believe that this innovative technique, along with future developments and upscaling, will be able to deliver online data for farmers on the availability of soil nitrate for their growing crops. By having real-time information on nitrate concentrations in the soil, farmers can accurately adjust fertilizer-application regimes according to the plants' needs in their concurrent growing phase to maximize yields and reduce the potential for groundwater contamination by nitrate.





**Authors contribution:**

E. Yeshno had conducted the experiment, analyzed the data and wrote most of this paper. S. Arnon
had assisted in developing the monitoring system at the electrical and optical engineering levels.
O.dahan had helped with designing the experimental concept and setup while having a major
contribution to the writing process and data analyses.

**Acknowledgements**

This work was funded by the Markus Foundation and Israel Innovation Authority (KAMIN Framework). The
authors wish to express their great appreciation to Michael Kugel, who stood behind each and every technical
aspect of the project while providing outstanding solutions for laboratory and field experiments.





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

**Figures and captions:**

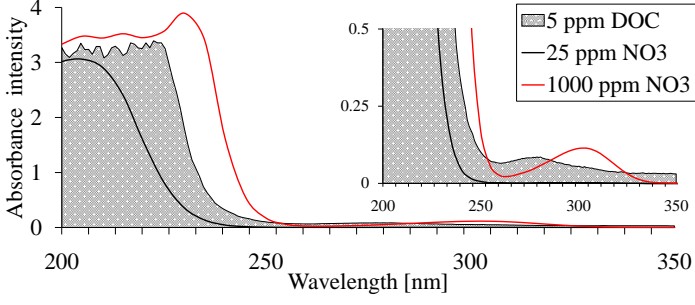

**Figure 1: Absorption spectra of nitrate at concentrations of 25 ppm, 1000 ppm, and 5 ppm dissolved organic carbon (DOC).**





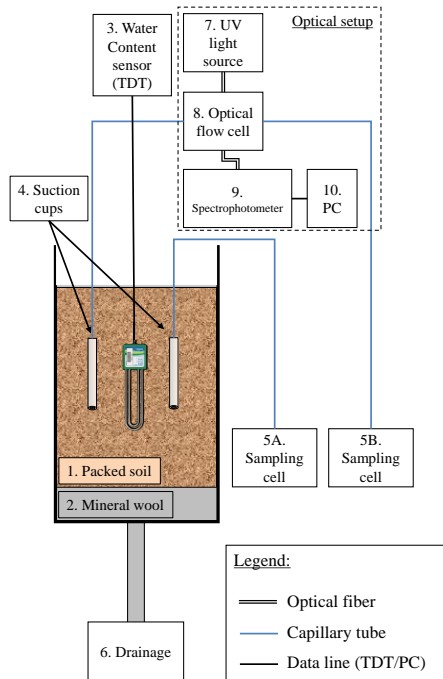

**Figure 2: Soil-packed column and optical setup for nitrate breakthrough curve experiment.**





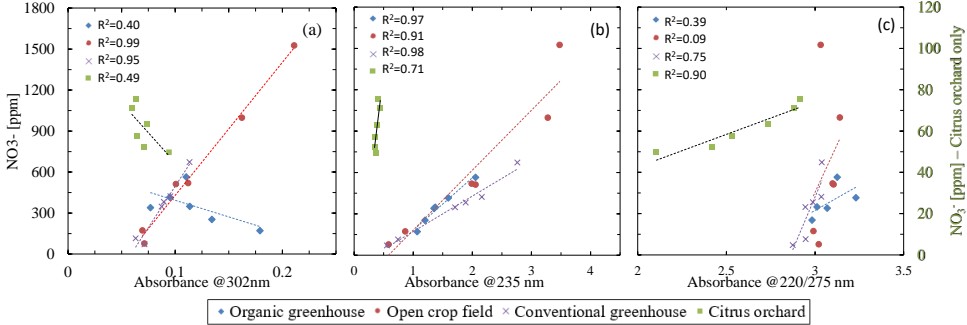

**Figure 3: Nitrate concentration vs. absorbance intensity at various wavelengths. Right ordinate presents nitrate concentration for the citrus orchard only.**

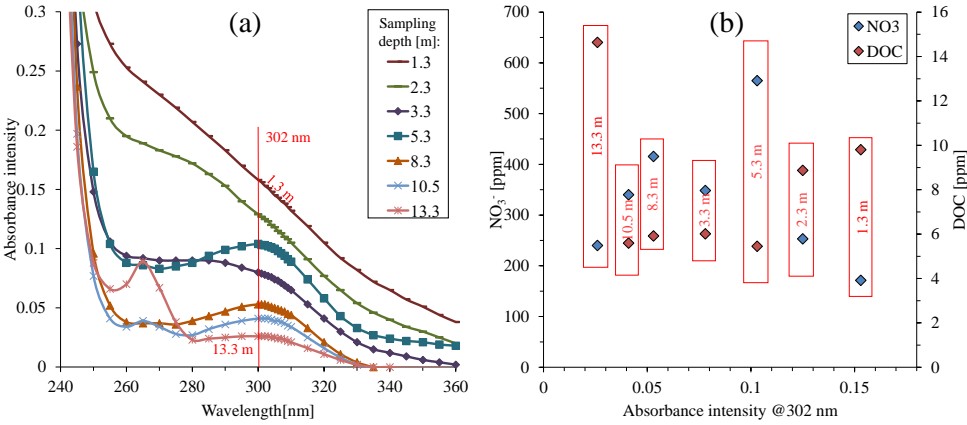

**Figure 4: Absorbance intensity in the 300 nm region of samples taken under the organic greenhouse. Both nitrate and dissolved organic carbon (DOC) concentration values are presented.**





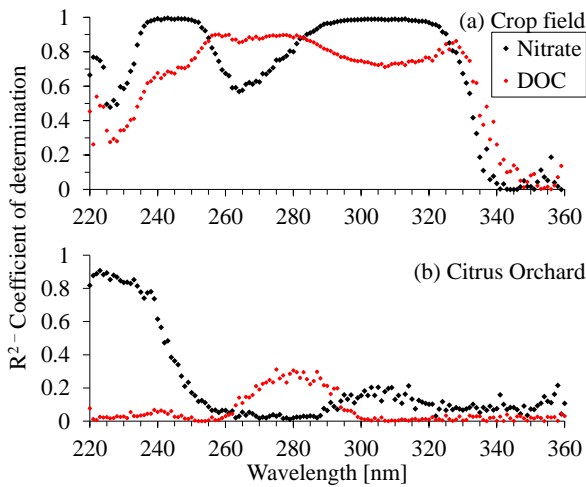

**Figure 5: Coefficient of determination ($R^2$) for nitrate and dissolved organic carbon (DOC) plotted against wavelength in the UV region for (a) crop field station and (b) citrus orchard.**

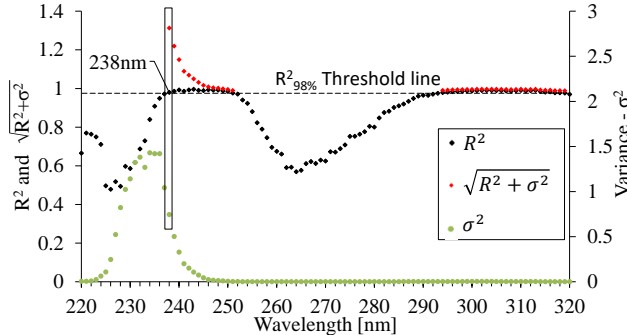

**Figure 6: Relationship between coefficient of determination ($R^2$), variance ($\sigma^2$) and the UV spectrum for the open crop field. $\sqrt{R^2 + \sigma^2}$ was calculated only for values where $R^2$ exceeded the set threshold at $R^2_{98\%}$. The maximum calculated value was determined as the optimal wavelength and was set to 238 nm.**





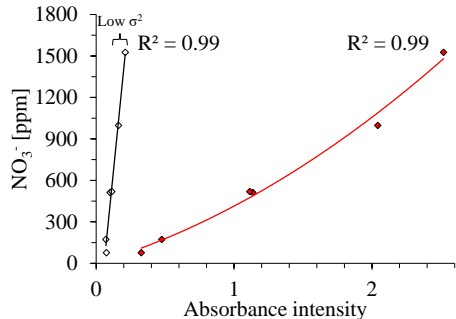

**Figure 7: Calibration curves created using absorbance data at 238 nm and 300 nm.**

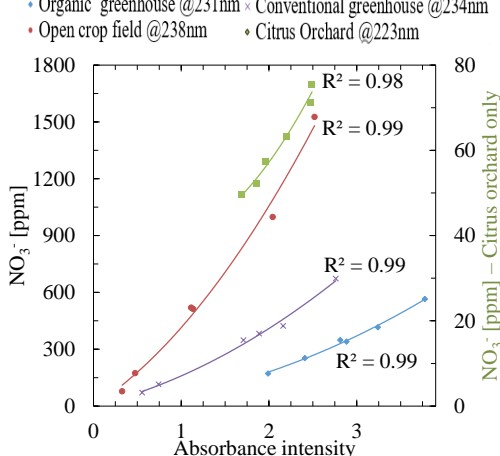

**Figure 8: Calibration equations for the four study sites. As can be seen on the chart legend, each of the sites has its own unique optimal wavelength for estimating nitrate concentration. Note that the right ordinate shows a lower concentration range than the left ordinate and is associated only with the citrus orchard.**






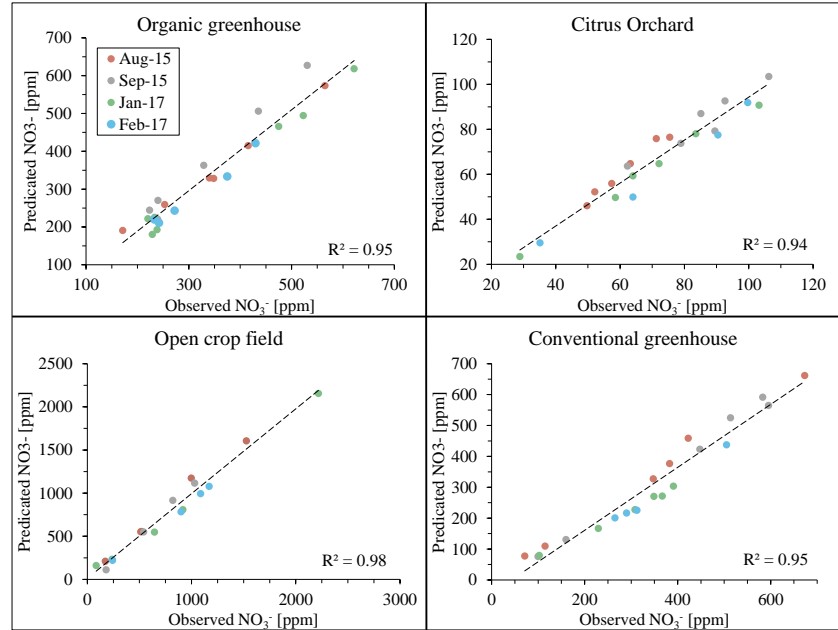

**Figure 9: Evaluation of nitrate concentration at the four study sites between the years 2015 and 2017.**





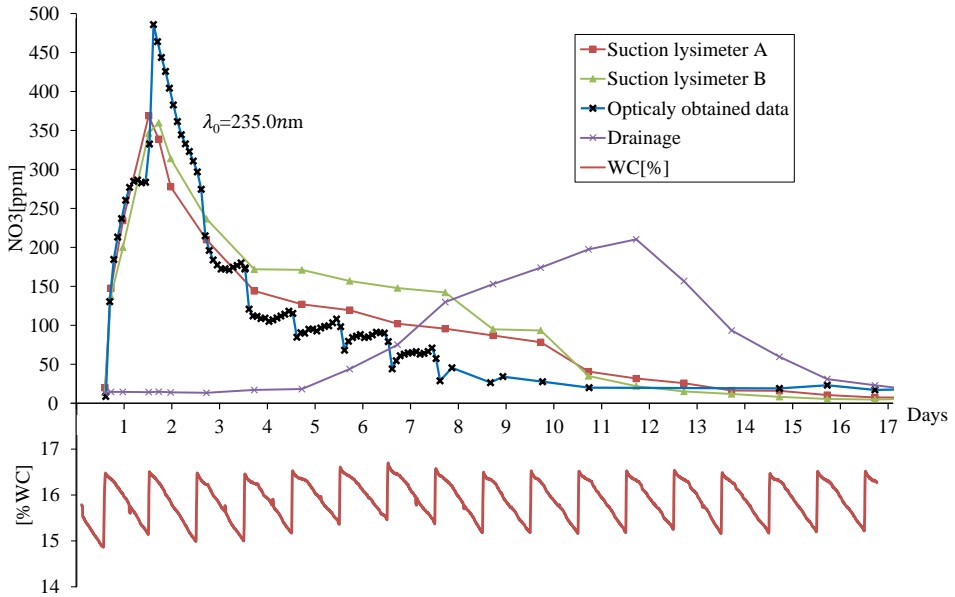


**Figure 10: Breakthrough curves plotted for physically sampled solution and calculated nitrate concentration, as obtained automatically by the optical setup. The bottom curve shows the soil water content as obtained by the water-content sensor (TDT).**

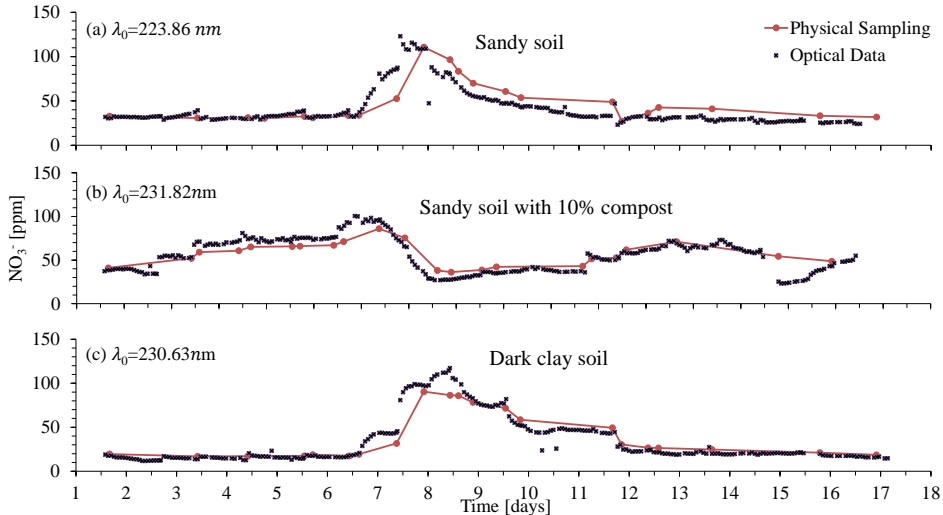

**Figure 11: Nitrate breakthrough curves for (a) sandy soil, (b) sandy soil with 10% compost and (c) dark clay soil.**



**Table 1: Nitrate concentration vectors obtained by ion chromatography for the conventional greenhouse porewater samples, along with their corresponding absorbance intensity vectors at different wavelengths. The $R^2$ column shows the correlation strength between the two vectors.**

| Wavelength [nm] | Nitrate concentration vectors [ppm] | | | | | | $R^2$ |
| | 849 | 657 | 650 | 857 | 121 | 212 | |
| | Absorbance intensity vectors | | | | | | |
| --- | --- | --- | --- | --- | --- | --- | --- |
| 190 | 2.381 | 2.274 | 2.274 | 2.334 | 2.325 | 2.245 | 0.216 |
| 195 | 3.122 | 3.146 | 3.093 | 3.148 | 3.043 | 3.076 | 0.770 |
| 200 | 3.289 | 3.284 | 3.352 | 3.343 | 3.231 | 3.205 | 0.666 |
| 230 | 3.764 | 3.591 | 3.695 | 3.797 | 1.515 | 2.371 | 0.916 |
| 235 | 2.659 | 2.869 | 2.365 | 2.896 | 0.612 | 0.935 | 0.930 |
| 237 | 1.864 | 2.103 | 1.634 | 2.072 | 0.424 | 0.633 | 0.909 |
