# Peer review of "Real time monitoring of nitrate in soils as a key for optimization of agricultural productivity and prevention of groundwater pollution"

_Hydrology and Earth System Sciences, 2019_

## Referee Comment (RC1) · Anonymous Referee #1 · 10 Jun 2019

June 10, 2019

A review of: Continuous in-situ monitoring nitrate concentration in soils – a key for groundwater protection from nitrate pollution. By Yeshno et al.

Summary and recommendation

This manuscript describes the development and first successful implementations of a new apparatus for real-time monitoring of soil-pore-water nitrate concentration. The apparatus is based on a customized suction cup and tubing with a small "dead volume" that can hold continuous flow to an optical flow-cell in which the pore-water are exposed to a UV lamp and absorbance is measured by a spectrophotometer, the pore-

water continue to flow nfrom the optical cell to a sample collector for lab analysis and calibration. A site-specific optimization of the working wave-length which consider the interference with dissolved organic carbon (DOC), the sensitivity of the absorbance to nitrate concentration and the correlation between them, is described. Site specific calibration was validated with samples from the same in-situ suction cups at 3 occasions within 2 years after the calibration samples. Monitoring nitrate in controlled tracer experiments in columns with different soils was shown to be comparable to nitrate lab-analysis of corresponding pore water samples. The suggested apparatus looks as a large step towards monitoring root zone nitrate / controlled N fertilization systems in agricultural fields. These type of systems can enhance N fertilization efficiency and reduce nitrate leaching to groundwater, significantly (nitrate is the no. 1 contaminant leading to disconnection of wells from direct supply to drinking water systems). Therefore I defiantly recommend publication in HESS after some clarifications and modifications listed below.

General comments

1) Continuous suction of pore-water from unsaturated porous medium (and bringing it up to surface in small diameter tubing) must impose some limitations of minimum water-content (soil-texture dependent) in which this apparatus can work (what suction pressures are imposed on the cups?). A TDT for water content monitoring was installed in the experimental setup, therefore I am sure the authors have some understandings considering the soil moisture conditions effects on the nitrate monitoring possibilities that are of interest for the HESS readership. Therefore I encourage the authors to elaborate on this issue. 2) The use of the term "absorbance intensity" throughout the text, figures and supplements instead of absorbance is somewhat inadequate. Absorbance is defined as the log of a ratio of light intensities. Change throughout.

Specific comments

1)L 20 "untreated" do you mean non-disturbed? 2)L 40 I would suggest to enhance the

arguments for this type of monitoring saying: Nitrate uptake was observed and modeled as passive uptake with a threshold root-zone concentration (Cmax) from which the roots can up take only S*Cmax (S - root water uptake, e.g. Simunek and Hopmans, 2009 (Ecological Modeling)). This mechanism imposes a jump in deep leaching of nitrate at times when C>Cmax, hence monitoring of nitrate concentration can serve as controller leading to increasing N use efficiency and decreasing groundwater contaminations. Values of Cmax for different crops were reported between 20 to 45 mg/l NO3-N, (Kurtzman et al., 2013; Levy et al., 2017 (HESS)).

3)L 90 – "second derivative spectroscopy" is not a clear phrase for most of the hydrology readership. 1-2 sentences defining this term will help. 4) L 200 – With small sample size (4-7 points) it would help to add to the R2 values also the P values of the slopes of the regression models, to enhance the sense of their significance. 5) L238, L240 "1.3 m" should be 1.3 m below surface or a depth of 1.3 m. Same for 13.3 m. 6) L310. I would start this paragraph with something like : A high R2 can be achieved also with wavelengths in which the sensitivity of the absorbance to nitrate concentration is extremely high and absorbance could not be used for estimating nitrate concentrations. Therefore the variance of the absorbance values that correlate well with the range of nitrate concentrations is a second criteria for choosing the best wavelength. Starting the paragraph with "Variance.." is ambiguous. 7) Figure 8 or L 337. Where are the calibration equations? Put them in the text or on the Figure. 8) L 417 delete "-based". 9) Figure 9. It would be better not to use the calibration data (red points) in this analysis, and draw the predicted-observed regression lines (and R2) only for the validation points of the 3 later sampling dates. That would give a better estimation of the performance of the method.

Please also note the supplement to this comment:
https://www.hydrol-earth-syst-sci-discuss.net/hess-2019-198/hess-2019-198-RC1-supplement.pdf

---

## Referee Comment (RC2) · Anonymous Referee #2 · 15 Jul 2019

Dear editor I appreciate the opportunity to review the manuscript entitled: 'Continuous in-situ monitoring of nitrate concentration in soils– a key for groundwater protection from nitrate pollution'. The authors present a new monitoring concept which is based on the application of UV absorption techniques to continuously observe nitrate concentrations in soil under various types of agriculture cultivation. This method is unique and might change our ability to trace nitrate in soils. Therefore, I recommend publishing this manuscript. Before publication of the manuscript, I suggest to re-organize the text (see specific comments below). The authors have the tendency to elaborate methods and techniques at irrelevant sections. Please, try to be more concise, it would help the reader to follow the manuscript.

[Figure]

Specific comments:

Line 1: Why limit the presented method to groundwater protection? Also agriculture management could benefit from knowing the amount of leached nitrate. I suggest the following title: 'Real time monitoring of nitrate in soils'

Lines 11-12: 'Rising nitrate. . .' – delete this sentence, you already mentioned the problems arising from overuse of nitrate.

Line 12: I suggest 'The development of . . .'

Line 22: delete 'the' and add 's' to soils

Lines 22 – 26: 'The system . . .' delete. The abstract should be concise.

Lines 28-34: I suggest to include two main challenges with nitrate fertilizer application. The first problem, as was mentioned in the referred lines, is the water resources pollution by nitrate. Note that the references related only to groundwater resources. You should indicate that there are other water resources, such as rivers, which are affected by nitrate. The other issue is agriculture management. For example, the method can help the farmer to time the nitrate fertilization application. You should indicate the challenges that agriculture management is facing with regards to nitrate application, just mention it concisely in a couple of lines. Delete the sentence regarding the Israeli problem, 'In Israel. . .'. You want to generalize your contribution.

Lines 43-46: I suggest moving these lines to the first paragraph.

Lines 48-51: I don't see the contribution of these lines to the introduction. You already mentioned the disadvantages in lack of real time monitoring of nitrate. I suggest deleting these lines.

Line 52: Today – delete

Lines 54-59: 'However, . . . solution' – I suggest deleting these lines. 'these methods are designed for water samples. . .' move to last paragraph, because the current manuscript

describes the next step or evolution of these methods.

Lines 65-67: delete. Put the references in line 65 after the words '...decades'.

Lines 67-98: I would move these lines to the method section. Here you talk about the challenges of applying the system at different environments. Or maybe make one paragraph, to explore the different implementations of the UV method.

Lines 178-194 – these are method descriptions. Please move these lines to the method section.

Line 196: '...between the two' – which two, there are three wavelengths.

Lines 226-228: this sentence is a bit vague. Do you want to say that the results indicate that different soil types should have different calibrations curves. Or in short, the soil texture might affect nitrate concentration readings using the UV method?

Lines 230-236: These lines contain a mix of discussion and information, which makes it hard to follow. You can start this paragraph from 'The absorption spectrum ...' and delete the preceding lines.

Line 242: 'it could be concluded' - the conclusions section is at the end of the manuscript. Please rephrase the sentence.

Lines 320-325: move these lines to the method section.

Lines 339-342: 'As ... equations' delete these lines.

Lines 342-345: these lines should be in methods section.

Lines 346-348: delete these lines. You mentioned the sampling dates earlier and if not do so.

Lines 348-349: the sentence is vague, Do you mean that the August 2015 was used as a reference curve?

Line 351: you use the word 'accordingly' far too many times. Delete 'Accordingly', and

write 'It suggests. . .'

Lines 354: 'be concluded' - the conclusions section is at the end of the manuscript. Please rephrase the sentence.

Lines 359-371: move to methods section.

---

## Author Comment (AC1) · 29 Jul 2019

We thank the reviewer for the encouraging statement on the importance of real time monitoring of nitrate to improve agricultural productivity while reducing water resources pollution potential.

General comments Comment 1: Continuous suction of pore-water from unsaturated porous medium (and bringing it up to surface in small diameter tubing) must impose some limitations of minimum water-content (soil-texture dependent) in which this apparatus can work (what suction pressures are imposed on the cups?). A TDT for water content monitoring was installed in the experimental setup, therefore I am sure the authors have some understandings considering the soil moisture conditions effects on the nitrate monitoring possibilities that are of interest for the HESS readership. Therefore, I encourage the authors to elaborate on this issue.

Reply to general comment 1: The efficiency of the nitrate monitoring system is indeed depended on its ability to extract a stream of soils solution, from the soil pores and into the sensor's optical flow cell. Accordingly, the system operation effectivity is depended upon the soil water potential. However, in agricultural soils, where the system is designed and intended to be installed, the water content is usually high enough to allow root uptake, and as such is sufficient enough to enable efficient operation of the monitoring system. Nevertheless, at low water content (water potential), as may happen between growing seasons or during dry periods, both water flow and nitrate transport is decrease dramatically, and consequently the potential for nitrate leachate out of the root zone to deep unsaturated zone is limited, thus reducing groundwater pollution potential. During the column experiments the hydraulic and suction parameter were set to represented typical agricultural soils condition for sandy loam, and were set to water content levels between 15 – 16.5 %. The porewater suction pressures levels were set between 600 - 800 mbar while typically the soil water pressure in these water contents is in the range of 830 - 950 mbar (figure 1) (Filipović et al., 2016). The manuscript was revised accordingly to account for the soil water potential impact on the measurement efficiency (lines 181 - 186). The sampling tube diameter (1.9 mm) and length (<10 m) did not impose limitation to pore-water stream from the porous interface to the optical cell.

Comment 2: The use of the term "absorbance intensity" throughout the text, figures and supplements instead of absorbance is somewhat inadequate. Absorbance is defined as the log of a ratio of light intensities. Change throughout.

Reply to general comment 2: The comment is accepted, and the manuscript was revised accordingly in all relevant places.

Specific comments:

Comment 1: L 20 "untreated" do you mean non-disturbed?

Reply to specific comment 1: The term "untreated" in line 20 is referring to soil solution and is mentioned to emphasize that the sampled soil solution was not filtered, diluted or spiked with reagents. For clarification the manuscript was revised and corrected accordingly (line 21).

Comment 2: L 40 I would suggest to enhance the arguments for this type of monitoring saying: Nitrate uptake was observed and modeled as passive uptake with a threshold root-zone concentration (Cmax) from which the roots can up take only S*Cmax (S - root water uptake, e.g. Simunek and Hopmans, 2009 (Ecological Modeling)). This mechanism imposes a jump in deep leaching of nitrate at times when C>Cmax, hence monitoring of nitrate concentration can serve as controller leading to increasing N use efficiency and decreasing groundwater contaminations. Values of Cmax for different crops were reported between 20 to 45 mg/l NO3-N, (Kurtzman et al., 2013; Levy et al., 2017 (HESS)).

Reply to comment 2: Comment is accepted, and a summary of the review's suggestion had been added (lines 39 and 47).

Comment 3: L 90 – "second derivative spectroscopy" is not a clear phrase for most of the hydrology readership. 1-2 sentences defining this term will help.

Reply to comment 3: Comment is accepted, and the manuscript was enhanced to better describe the second derivative spectroscopy technique (Lines 92 – 96).

Comment 4: L 200 – With small sample size (4-7 points) it would help to add to the R2 values also the P values of the slopes of the regression models, to enhance the sense of their significance.

Reply to comment 4: Comment is accepted, the P-values for each curve was added in the body of figure 3. Additionally, the methods section was revised to account for

the R2 and their corresponding P values analyses MATLAB liner regression fitting tool (Lines 157 – 159).

Comment 5: L 238, L 240 "1.3 m" should be 1.3 m below surface or a depth of 1.3 m. Same for 13.3 m.

Reply to comment 5: the comment is accepted, and the manuscript had been corrected accordingly, lines 219, 233 and 236.

Comment 6: L 310. I would start this paragraph with something like: A high R2 can be achieved also with wavelengths in which the sensitivity of the absorbance to nitrate concentration is extremely high, and absorbance could not be used for estimating nitrate concentrations. Therefore, the variance of the absorbance values that correlate well with the range of nitrate concentrations is a second criteria for choosing the best wavelength. Starting the paragraph with "Variance.." is ambiguous.

Reply to comment 6: The comment is accepted and the suggested comment by the reviews had been addend to the manuscript (lines: 305 – 308).

Comment 7: Figure 8 or L 337. Where are the calibration equations? Put them in the text or on the Figure.

Reply to comment 7: The comment is accepted and the corresponding equation for each curve had been added to figure 8.

Comment 8: L 417 delete "-based".

Reply to comment 8: accepted, the word "based" had been deleted from the manuscript (line 402).

Comment 9: Figure 9. It would be better not to use the calibration data (red points) in this analysis, and draw the predicted-observed regression lines (and R2) only for the validation points of the 3 later sampling dates. That would give a better estimation of the performance of the method.

Reply to comment 9: comment accepted, the data points from August – 2015 was removed from the plot and the regression lines are now account for the data of the remaining 3 sampling dates.

References:

Filipović, V., Ondrasek, G. and Filipović, L.: Modelling Water Dynamics, Transport Processes and Biogeochemical Reactions in Soil Vadose Zone., 2016.

Please also note the supplement to this comment:
https://www.hydrol-earth-syst-sci-discuss.net/hess-2019-198/hess-2019-198-AC1-supplement.pdf

───────────────────────────

Figure 1 – Loam, sand and clay soil's retention curves (Filipovic et al., 2016). The dashed lines mark the matric potential for Loam at about 15 % water content, and which is about 900 cm (equivalent to ~880 mbar).

**Fig. 1.**

---

## Author Comment (AC2) · 29 Jul 2019

We wish to thank the reviewer for its encouraging statement "This method is unique and might change our ability to trace nitrate in soils". This is definitely the driving force for our research endeavor.

General comment: The authors have the tendency to elaborate methods and techniques at irrelevant sections. Please, try to be more concise, it would help the reader to follow the manuscript.

Reply to general comment: We accept the comment and revised the manuscript in

several places following the specific comments.

Specific comments: Comment 1: L1 Why limit the presented method to groundwater protection? Also agriculture management could benefit from knowing the amount of leached nitrate. I suggest the following title: 'Real-time monitoring of nitrate in soils. Reply to comment 1: We agree with the comment and believe that the benefit to the agricultural sector is as important as protecting groundwater from pollution hazard. As such we take the reviewer advice and revised the title to include both the agricultural and environmental aspects. Accordingly, the new title is: "Real-time monitoring of nitrate in soils as a key for optimization of agricultural productivity and prevent groundwater pollution".

Comment 2: L 11 - 12 'Rising nitrate. . .' – delete this sentence, you already mentioned the problems arising from overuse of nitrate. Reply to comment 2: It is truth that overuse of fertilizers in agriculture may end in down leaching of nitrate to groundwater (previous sentence). Nevertheless, the sentence "Rising nitrate concentration in aquifers is recognized as a worldwide environmental problem that contributes to water scarcity" give a hint to the severity of the phenomenon as a global threat to water resources in general, and particularly to groundwater. Therefore, although part of it is somewhat redundant to professional eyes, we believe that in this particular case elaboration is important even if it include a short repletion.

Comment 3: L 12 I suggest 'The development of . . .' Line 22: delete 'the' and add 's' to soils Reply to comment 3: Comment accepted, and the text was revised accordingly (line 13).

Comment 4: L 22 – 26: 'The system . . .' delete. The abstract should be concise. Reply to comment 4: The last two sentences of the abstract provide the reader with three very important aspects: (1) scientific- impact of irrigation pattern on nitrate mobility, (2) Agricultural yield - optimization of fertilizers application for improvement of field production, and (3) Water resources protection. Nevertheless, we agree with the reviewer

that a concise abstract is essential, and we revised these sentences to improve clarity while keeping notion of the three aspects. "Real-time, high-resolution measurement on nitrate concentration in the soil revealed the complex variations in soil nitrate concentrations in response to fertigation pattern. Such data is crucial for optimizing fertilizer application and reduce the pollution potential of groundwater." (Line 25).

Comment 5: L 28 – 34 I suggest to include two main challenges with nitrate fertilizer application. The first problem, as was mentioned in the referred lines, is the water resources pollution by nitrate. Note that the references related only to groundwater resources. You should indicate that there are other water resources, such as rivers, which are affected by nitrate. The other issue is agriculture management. For example, the method can help the farmer to time the nitrate fertilization application. You should indicate the challenges that agriculture management is facing with regards to nitrate application, just mention it concisely in a couple of lines. Delete the sentence regarding the Israeli problem, 'In Israel. . .'. You want to generalize your contribution. Reply to comment 5: We agree to the reviewer's comment that nitrate pollution is not only in concern to groundwater resources and the manuscript is revised to account for surface water (lines 29 - 31). Additionally, the section regarding nitrate pollution of groundwater in Israel has been deleted. Finally, the possible gain from nitrate monitoring in soils in regards to agriculture management and fertilizer application timing had been added to the manuscript (lines 44 - 46).

Comment 6: L 43 - 46: I suggest moving these lines to the first paragraph. Reply to comment 6: Although both lines 51 - 54 and the first paragraph (lines 28 – 34) are dealing with groundwater pollution by nitrate, the first paragraph focuses mainly on the global problem of water pollution by nitrate, while lines 51 - 54 mainly attends on the limitation of the available technology to deal with real-time variability of nitrate in soils.

Comment 7: L 48 – 51 I don't see the contribution of these lines to the introduction. You already mentioned the disadvantages in lack of real-time monitoring of nitrate. I suggest deleting these lines. Reply to comment 7: The comment is accepted, and the

related text was deleted from the manuscript (lines 56 – 59).

Comment 8: L 52 Today – delete Reply to comment 8: The comment is accepted, and the text was revised accordingly (line 57).

Comment 9: L 54 - 59 'However, solution' I suggest deleting these lines. 'these methods are designed for water samples.' move to last paragraph, because the current manuscript describes the next step or evolution of these methods. Reply to comment 9: The comment is accepted, and the text was revised accordingly (line 59).

Comment 10: L 65 – 67 delete. Put the references in line 65 after the words '. . .decades'. Reply to comment 10: The comment is accepted the manuscript was revised (lines 68 – 69).

Comment 11: L 67 – 98 I would move these lines to the method section. Here you talk about the challenges of applying the system at different environments. Or maybe make one paragraph, to explore the different implementations of the UV method. Reply to comment 11: The manuscript in the mentioned lines is divided into three main sections. The first section (lines 71 - 80), brings a short description of past work showing the potential to use UV absorption spectroscopy to perform in-situ analyses of nitrate in soils. The second section (lines 81 - 88) shows a limitation which can occur when trying to apply UV absorption spectroscopy technique to measure nitrate in soil's porewater when Dissolved Organic Carbon (DOC) is present. The third section (lines 89 - 102) is a short review of studies that were trying to deal with the problem of measuring aqueous nitrate in the presence of DOC by few spectral techniques. At the course of this study we have tried, unsuccessfully to apply these methods for measuring nitrate in porewater samples from few agricultural sites, and with the presence of DOC. For this reason, this research was focusing on finding a robust method that would enable the use of UV absorption technique to monitor nitrate in soils, at high time resolution and with the presence of DOC. Therefore, the mentioned text (lines 71 – 102) is described as few case studies during the introduction section and not under the methodology
section. We believe that this section improve clarity and essential as scientific background.

Comment 12: L 178 - 194 These are method descriptions. Please move these lines to the method section. Reply to comment 12: The comment is accepted. A similar text appears in the methods section (lines 118 – 129) and therefore the current text has been removed from the manuscript.

Comment 13: L 196: '. . .between the two' – which two, there are three wavelengths. Reply to comment 13: The comment is accepted, and the text was clarified at lines 196 - 197.

Comment 14: L 226 – 228 This sentence is a bit vague. Do you want to say that the results indicate that different soil types should have different calibrations curves? Or in short, the soil texture might affect nitrate concentration readings using the UV method? Reply to comment 14: The spectral analysis shows that different sites may have appropriate calibration curve for nitrate concentrations at different wavelengths, which implies the possibility of adopting a unique wavelength for each site. The manuscript was revised for clarified (lines 228 – 230).

Comment 15: L 230 – 236 These lines contain a mix of discussion and information, which makes it hard to follow. You can start this paragraph from 'The absorption spectrum . . .' and delete the preceding lines. Reply to comment 15: The reviewer comment is accepted, and the text was revised accordingly (line 232).

Comment 16: L 242 'it could be concluded' - the conclusions section is at the end of the manuscript. Please rephrase the sentence. Reply to comment 16: The comment is accepted, and the text was revised at line 238.

Comment 17: L 320 – 325 move these lines to the method section. Lines 339-342: 'As . . . equations' delete these lines. Reply to comment 17: Lines 319 - 324 are describing the last phase of an algorithm developed to overcome the difficulties of measuring

nitrate in porewater samples containing DOC. Since the development of the algorithm is an outcome of this research, and it is not a standard analysis, it is brought under the results and discussion section rather than the materials and methods. Additionally, lines 338 – 343 (previously 339 – 342) had been revised to gain a more concise manuscript.

Comment 18: L 342 – 345 these lines should be in methods section. Reply to comment 18: Lines 339 – 342 describes a test conducted to insure the previously described algorithm measurement accuracy and thus relevancy during long operations when the chemistry of the soil solution in the field may change in response to different seasonal events (temperature changes, different irrigation scheme, different crops, etc.). Since the understanding that variations in the solution chemical composition can lead to measurement drifts when applying UV absorption spectroscopy techniques, is one of the findings of this research, this chapter appears following the description of the newly developed algorithm at the results and discussion section.

Comment 19: L 346- 348: delete these lines. You mentioned the sampling dates earlier and if not do so. Reply to comment 19: Comment is accepted. The manuscript of chapter "3.5. Stability and consistency of the specific calibration curves" been revised as elaborated under the replies for comments 18 and 20.

Comment 20: L 348 – 349 The sentence is vague, Do you mean that the August 2015 was used as a reference curve? Reply to comment 20: The data collected from august 2015 samples were used as input for the site-specific algorithm, and as the algorithm output, a calibration equation at different wavelength were obtained for each field site. The stability of these calibration equations had been tested on samples from additional sampling campaigns later in 2015, and at 2017, where results from standard laboratory analyses (observed nitrate concentrations) were plotted in reference to the result of the calibration equation, obtained at august 2015 (predicate nitrate concentration). To improve clarity the manuscript is revised (lines 343 - 348).

Comment 21: L 351 You use the word 'accordingly' far too many times. Delete 'Accordingly', and write 'It suggests. . .' Reply to comment 21: The comment is accepted, and text has been revised accordingly (line 348)

Comment 22: L 354 'be concluded' - the conclusions section is at the end of the manuscript. Please rephrase the sentence Reply to comment 22: The comment is accepted, and text has been revised accordingly (line 351)

Comment 23: L 359 - 371 Move to methods section Reply to comment 23: A similar text appears in the methods section (lines 176 – 192) and therefore the current text has been removed from the manuscript.

Please also note the supplement to this comment:
https://www.hydrol-earth-syst-sci-discuss.net/hess-2019-198/hess-2019-198-AC2-supplement.pdf